# NECTIN4: A Novel Therapeutic Target for Melanoma

**DOI:** 10.3390/ijms22020976

**Published:** 2021-01-19

**Authors:** Yuka Tanaka, Maho Murata, Che-Hung Shen, Masutaka Furue, Takamichi Ito

**Affiliations:** 1Department of Dermatology, Graduate School of Medical Sciences, Kyushu University, Fukuoka 812-8582, Japan; yukat53@med.kyushu-u.ac.jp (Y.T.); muratama@dermatol.med.kyushu-u.ac.jp (M.M.); furue@dermatol.med.kyushu-u.ac.jp (M.F.); 2National Institute of Cancer Research, National Health Research Institutes, Tainan 70456, Taiwan; chshen@nhri.org.tw; 3Research and Clinical Center for Yusho and Dioxin, Kyushu University Hospital, Fukuoka 812-8582, Japan

**Keywords:** NECTIN4, malignant melanoma, antibody–drug conjugate, BRAF, BRAFi resistance

## Abstract

Malignant melanoma is the most common lethal skin cancer and causes death in a short time when metastasized. Although BRAF inhibitors (BRAFi) have greatly improved the prognosis of BRAF-mutated melanoma, drug resistance is a major concern even when they are combined with MEK inhibitors. Alternative treatments for BRAFi-resistant melanoma are highly anticipated. Nectin cell adhesion molecule 4 (NECTIN4) is highly expressed and associated with progression in tumors. We aimed to investigate the role of NECTIN4 in melanoma and its potency as a therapeutic target using 126 melanoma samples and BRAFi-resistant cells. Immunohistochemically, most of the clinical samples expressed NECTIN4, at least in part. NECTIN4 was highly expressed in BRAF-mutated melanoma and its high expression was associated with disease-free survival. In BRAFi-resistant melanoma cells, NECTIN4 and the PI3K/Akt pathway were upregulated, along with the acquisition of BRAFi resistance. Monomethyl auristatin E, a cytotoxic part of NECTIN4-targeted antibody–drug conjugate, was effective for BRAF-mutated or BRAFi-resistant melanoma cells. NECTIN4 inhibition increased the sensitivity of BRAFi-resistant cells to BRAFi and induced apoptosis. In conclusion, we revealed the expression and roles of NECTIN4 in melanoma. Targeted therapies against NECTIN4 can be a novel treatment strategy for melanoma, even after the acquisition of BRAFi resistance.

## 1. Introduction

Malignant melanoma is one of the most common lethal skin cancers, arising from melanocytes. In recent decades, the incidence of melanoma has been increasing worldwide and the estimated annual increase in its incidence is approximately 3–7% for Caucasians [1,2,3]. Most melanomas are cutaneous types and frequently occur in sunlight-exposed areas, since ultraviolet exposure is a known risk factor [1,2]. Although localized melanomas can be treated by surgical excision (5-year survival rate > 90%), melanoma frequently becomes invasive and metastasizes. Dacarbazine-based chemotherapy or radiation therapy used to be applied for unresectable melanoma, but the efficacy was low (survival rate < 10%) [4,5]. In recent years, immune checkpoint inhibitors (ICI), such as PD-1 inhibitors (nivolumab, pembrolizumab) and a CTLA4 inhibitor (ipilimumab), have been applied for the treatment of advanced melanoma, which dramatically improved the prognosis of patients [6,7]. However, there is still a subset of patients with primary/acquired drug resistance [8,9], indicating the urgent need for alternative treatments.

As is usually observed in cancers, gene mutations have been found in melanoma, such as in *BRAF*, *KIT*, *NF*, *NRAS*, and *PTEN* [10]. In particular, the mutational activation of BRAF is observed at a high frequency (about 50%) and BRAF^V600E^ mutation covers over 90% of the BRAF mutations [11,12,13]. These mutations activate the MAPK pathway, further triggering aberrant cell proliferation, inhibiting apoptosis and thus promoting tumor progression [12,13]. Since BRAF mutation is frequently observed in melanoma, BRAF inhibitors (BRAFi: dabrafenib, vemurafenib, and encorafenib) are now applied for clinical use, and show drastically higher clinical responses than dacarbazine-based treatments [13,14]. However, recurrence due to the acquisition of drug resistance is widely recognized as a limitation of BRAF-targeted therapy [15,16]. Although the use of BRAFi in combination with MEK inhibitors (MEKi) has partly solved this problem, resistance to BRAFi/MEKi still occurs. Thus, alternative or supportive treatments are eagerly anticipated.

Nectin cell adhesion molecule 4 (NECTIN4), a single-pass type I immunoglobulin-like membrane protein that mainly localizes in adherens junctions [17,18], belongs to the nectin family, which mediates various cell functions such as proliferation, differentiation, migration, and invasion [19,20,21]. NECTIN4 is overexpressed in many types of human cancers, such as urothelial cancer, and its abnormal expression is associated with tumor progression by increasing proliferation and angiogenesis, and decreasing apoptosis [22,23,24]. These suggest that NECTIN4 is a potential therapeutic target. Indeed, enfortumab vedotin, a kind of antibody–drug conjugates (ADCs) which targets NECTIN4, has been reported to suppress NECTIN4-overexpressing tumor cell growth and further evaluated preclinically in human breast, bladder, pancreas, lung, and urothelial cancers [24,25,26,27]. Moreover, several clinical trials are ongoing for evaluating its effects on patients with urothelial cancers expressing high NECTIN4 [28,29]. Thus, NECTIN4-targeted therapy may serve as a potent strategy for treating cancers with high NECTIN4 expression. In skin tissue, NECTIN4 is enriched in epidermal keratinocytes and skin appendages [30,31,32]. However, the association between NECTIN4 expression and the progression of melanoma and its function in melanoma is largely unknown. 

To investigate the NECTIN4 expression in melanoma, we first assessed NECTIN4 in melanoma patients. NECTIN4 was highly expressed in BRAF^V600E^-mutated melanoma, and its high expression was associated with poor prognosis. We further showed that NECTIN4 was upregulated in BRAFi-resistant melanoma cells compared with their BRAFi-sensitive counterparts. This work uncovers a link between NECTIN4 expression with BRAF^V600E^ mutation and with BRAFi resistance in melanoma, and provides a rationale for using NECTIN4-targeted drugs to treat melanoma with high NECTIN4.

## 2. Results

### 2.1. Characteristics of the Study Cohort

Comprehensive demographic and clinical data of 126 patients with primary malignant melanoma are shown in Table 1. The mean age was 63.7 years (range: 16–88) and 43.7% were male and 56.3% were female. The primary tumor site was predominantly acral areas (hand; 19.8%, foot; 58.7%), and the major histopathological subtype was acral lentiginous melanoma (74.6%). Ulceration was found in 41.3%. The proportion of VE1-positive cases was 30.2%.

### 2.2. Association of NECTIN4 with Clinicopathological Factors and Prognostic Impact in Melanoma

As expected, NECTIN4 was observed in the epidermis (in the cytoplasm and on the membrane) (Figure 1a). SOX10, a melanoma specific marker, was stained simultaneously to distinguish melanoma cells. Importantly, NECTIN4 was expressed in melanomas and co-expressed with SOX10 (Appendix A). Then, NECTIN4 expression in melanoma patients was examined and presented as the H-score (mean: 48.1, range: 0–225, median: 30, Appendix A) [33]. Samples were divided into two groups by the mean H-score: NECTIN4-low (H-score ≤ 48.1) and NECTIN4-high (H-score > 48.1). The associations between immunohistochemical NECTIN4 expression and clinicopathological factors was analyzed. NECTIN4 was high in 35.7% and low in 64.3% patients. Among the factors analyzed, only VE1 staining, which indicates the presence of BRAF^V600E^ mutation, was significantly positively associated with NECTIN4 (*p* = 0.0011, Table 2). Of note, patients with NECTIN4-high melanoma showed significantly poor disease-free survival (DFS) compared with those with NECTIN4-low melanoma (*p* = 0.0358) (Figure 1b). As for melanoma-specific survival (MSS) and overall survival (OS), NECTIN4-high patients tended to show shortened survival (*p* = 0.196 for MSS and *p* = 0.0733 for OS) (Figure 1c,d). 

### 2.3. NECTIN4 Is Upregulated in BRAFi-Resistant Melanoma Cells

We next investigated the association between NECTIN4 and BRAF mutation. The intensity of NECTIN4 was positively correlated with VE1 staining (Table 2, *p* = 0.0011). This implies that NECTIN4 might be involved in malignant transformation and drug-resistance development in BRAF^V600E^-melanomas. We then generated dabrafenib (a widely used BRAFi)-resistant (DR) melanoma cells in MEL1617 and SKMEL28 melanoma cell lines using a stepwise selection by increasing concentration of dabrafenib. Most MEL1617 melanoma cells originally presented a polygonal or stellate shape, whereas MEL1617-DR cells showed a flatter and epithelial-like shape (Figure 2a). Although the growth of MEL1617 cells was strongly inhibited by dabrafenib, MEL1617-DR cells could proliferate even in the presence of dabrafenib (Figure 2b). Similar results were obtained in SKMEL28 and SKMEL28-DR cell lines (Figure 2c,d). Besides, activation of EGFR and MEK/ERK signal, which are the well-known factors of BRAFi-resistance [34,35,36,37], was observed in MEL1617-DR and SKMEL28-DR cell lines (Appendix A). These results confirmed that MEL1617-DR and SKMEL28-DR cell lines have acquired drug resistance to dabrafenib. Next, the NECTIN4 expression was examined. NECTIN4 was expressed in these melanoma cells in both mRNA and protein levels (Figure 2e,f). NECTIN4 was significantly increased in MEL1617-DR and SKMEL28-DR compared with their drug-sensitive counterparts. These results show that NECTIN4 is expressed in melanoma cells and that its expression is upregulated in BRAFi-resistant cells.

Since NECTIN4 is expressed in melanoma cells, we next examined whether these NECTIN4-expressing melanoma cells are sensitive to MMAE, a cytotoxic part of enfortumab vedotin. The concentration of MMAE in patients’ peripheral blood is 2.6–5.0 ng/mL [38]. The cells were treated with various concentrations of MMAE and 1.25 nM of MMAE was enough to suppress cell growth at 48 h after treatment in all melanoma cell lines (Figure 2g,h). Considering the NECTIN4 expression in melanoma cells and their sensitivity to MMAE, enfortumab vedotin may act as an effective anti-tumor drug against melanoma, and BRAFi-resistant cells are more likely to be sensitive to it.

### 2.4. Akt Phosphorylation Is Enhanced in BRAFi-Resistant Melanoma Cells

NECTIN4 is known to activate the PI3K/Akt pathway, which regulates tumor cell proliferation and tumor growth [39,40]. Since NECTIN4 was upregulated in the BRAFi-resistant cells, we next investigated whether the PI3K/Akt pathway is also activated or not. The phosphorylation status of Akt, a downstream molecule of the PI3K, was assessed, and it was significantly upregulated in MEL1617-DR and SKMEL28-DR compared with their drug-sensitive counterparts (Figure 3a,b). Thus, the PI3K/Akt pathway is activated in these BRAFi-resistant cells, implying an interaction between NECTIN4 upregulation and PI3K activation in BRAFi resistance.

### 2.5. Knockdown of NECTIN4 Impairs PI3K/Akt Pathway and Increases Apoptosis of Melanoma Cells

To investigate the relationship between NECTIN4 and PI3K/Akt activation, we inhibited NECTIN4 and examined whether it alters the Akt phosphorylation. NECTIN4 was significantly downregulated by siRNA in all cell lines (Figure 3c,d). The Akt phosphorylation was significantly reduced by NECTIN4 knockdown in all cell lines compared with that in control siRNA-transfected cells (Figure 3e,f). These results indicate that NECTIN4 is involved in the PI3K/Akt activation in melanoma cells.

PI3K/Akt pathway activation increases the proliferation of cancer cells and prevents apoptosis. Since NECTIN4 and phosphorylated Akt were both increased in the BRAFi-resistant cells and NECTIN4 inhibition reduced the phosphorylated Akt, we speculated that NECTIN4 expression and the subsequent PI3K/Akt pathway activation may decrease the apoptosis of the cells. Then, we evaluated the apoptotic cells by the Annexin V-propidium iodide (PI) staining in NECTIN4-inhibited cells (Figure 4). The percentages of the Annexin V-positive PI-positive (late apoptotic) cells and Annexin V-positive PI-negative (early apoptotic) cells among the total cells were analyzed at 48 h after siRNA transfection (Figure 4a). The percentage of total apoptotic cells (late and early) was significantly increased in the NECTIN4-knockdown MEL1617 (*p* = 0.0318) and NECTIN4-knockdown MEL1617-DR (*p* = 0.0381) compared with control siRNA-treated condition (Figure 4b). The percentage of total apoptotic cells was also significantly increased in the NECTIN4-knockdown SKMEL28 (*p* = 0.00412) and NECTIN4-knockdown SKMEL28-DR (*p* = 0.00698) compared with control siRNA-treated condition (Figure 4c). These results indicate that NECTIN4 may be partially responsible for the regulation of apoptosis in melanoma cells.

### 2.6. PI3K/Akt and NECTIN4 Inhibition Improved the Sensitivity of Melanoma Cells to BRAFi

To investigate whether the inhibition of the PI3K/Akt pathway improved the sensitivity of melanoma cells to the BRAFi, cells were treated with LY294002, a PI3K inhibitor, in combination with dabrafenib and assessed for the cell viability. The inhibition of PI3K/Akt by LY294002 was confirmed by analyzing the pAkt/Akt ratio. At 48–72 h after LY294002 treatment, phosphorylation of Akt was significantly inhibited in all of the cell lines tested (Appendix A). The proliferation of MEL1617 was originally inhibited by dabrafenib and the additional treatment with LY294002 further suppressed the cell proliferation. MEL1617-DR, which proliferated in the presence of dabrafenib, showed significantly suppressed proliferation upon PI3K/Akt inhibition (Figure 5a). SKMEL28 and SKMEL28-DR showed similar results, LY294002 treatment significantly suppressed the proliferation of both cell lines when treated with dabrafenib (Figure 5b).

Next, the effects of NECTIN4 inhibition on the sensitivity of melanoma cells to BRAFi were examined. NECTIN4 expression and the phosphorylation of Akt were both significantly decreased by siRNA in all cell lines (Appendix A). In the MEL1617 and the MEL1617-DR cultured with dabrafenib, NECTIN4 inhibition slightly but significantly inhibited the proliferation of the cells compared with that of control siRNA-treated cells (Figure 5c). In the SKMEL28 and the SKMEL28-DR cultured with dabrafenib, NECTIN4 inhibition significantly inhibited the proliferation of the cells compared with that of control siRNA-treated cells (Figure 5d). 

Taking all results together, the NECTIN4 induction and the subsequent PI3K/Akt pathway activation may cause BRAFi resistance and promote melanoma cell proliferation, suggesting that NECTIN4-targeted therapy may serve as a potent treatment for patients with BRAFi-resistant melanoma.

### 2.7. NECTIN4 Inhibition Down-Regulated ERK Signal in BRAFi-Resistant Melanoma Cells

The activation of MEK/ERK signal is a well-known feature of BRAFi-resistance [34,35,36,37], and it was observed in our BRAFi-resistant cell lines as well (Appendix A). To obtain further insight into the relationship of NECTIN4 and BRAFi resistance, the status of the MEK/ERK pathway was assessed in NECTIN4-inhibited BRAFi-resistant melanoma cell lines. The phosphorylated ERK, but not phosphorylated MEK, was significantly decreased in NECTIN4-inhibited cells compared to control siRNA-transfected cells in both MEL1617-DR and SKMEL28-DR cell lines (Figure 6). These results imply that NECTIN4 partly contributes to the drug-resistance of these melanoma cell lines by affecting ERK signaling.

## 3. Discussion

In this paper, we report for the first time that NECTIN4 is expressed in human melanoma, with an especially higher frequency in BRAF-mutated melanoma. Immunohistochemically, high NECTIN4 expression was associated with worse prognosis of the patients. In vitro analyses revealed increased NECTIN4 in the BRAFi-resistant melanoma cells and upregulation of the PI3K/Akt pathway. Knockdown of NECTIN4 induced apoptosis and improved sensitivity to BRAFi, suggesting the roles of NECTIN4 in the survival of BRAFi-resistant melanomas.

The expression of NECTIN4 is generally low in normal tissues, whereas it increases in various malignant tumors; its association with tumor progression has been reported among breast, urothelial, pancreatic, lung, and ovarian cancers [22,23,24]. Patients with NECTIN4-high tumors had significantly poorer overall survival than those with low ones in pancreatic cancer [22]. In addition, NECTIN4 expression was reported to be positively correlated with Ki67 and the silencing of NECTIN4 inhibited the proliferation of human pancreatic cancer cells, implying its role in the proliferation of tumor cells [22]. The involvement of NECTIN4 in tumor progression has been reported in other tumors as well, including gallbladder, breast, gastric, and papillary thyroid cancers [39,40,41,42]. In normal skin, NECTIN4 is mainly expressed in the epidermal keratinocytes and skin appendages [30,31,32]; however, its expression in skin cancers has not been extensively investigated. In agreement with reports on other tumors, we found that NECTIN4 is expressed in melanoma and its expression reflects a poorer prognosis. We also showed that the silencing of NECTIN4 resulted in the reduction in the proliferation and induction of apoptosis in melanoma cells. Mutational activation of BRAF is associated with tumor progression and BRAF-mutated melanoma showed higher NECTIN4 expression than melanoma with wild-type BRAF. In the current study, we found that high NECTIN4 expression was significantly associated with worse DFS, but not with MSS and OS. In general, MSS and OS is influenced by the systemic treatments after metastasis, while DFS may reflect the primary tumor characteristics more accurately. BRAF-mutated melanoma can be treated with BRAFi and is more sensitive to ICI than BRAF-wild-type melanoma. Indeed, 13 patients in our cohort were treated with BRAFi and/or ICI; these treatments might have led to the failure to find a significant correlation with MSS and OS. NECTIN4 is likely to be associated with the tumor prognosis and plays roles in the progression of melanoma.

Regarding the mechanisms underlying NECTIN4-mediated tumor progression, previous reports demonstrated the involvement of the PI3K/Akt pathway in multiple cancers [39,40,41,42]. Higher NECTIN4 expression was associated with poorer prognosis of gastric cancer and papillary thyroid cancer, and the progression of cancer was promoted by NECTIN4 via activation of the PI3K/Akt pathway [41,42]. In addition to the PI3K/Akt pathway, BRAF and its downstream MEK/ERK pathway regulates the growth of melanoma cells [34]. However, a study on gallbladder cancer failed to show the involvement of the MEK/ERK pathway in the oncogenic function of NECTIN4, despite the positive correlation of the PI3K/Akt pathway [39]. Consistent with these reports, our results showed that the PI3K/Akt signaling was activated in BRAFi-resistant melanoma cells, where NECTIN4 levels were increased. Silencing of NECTIN4 resulted in the inhibition of the PI3K/Akt pathway, followed by decreased proliferation and increased apoptosis. The inhibition of PI3K/Akt by LY294002 also prevented the proliferation of the cells and improved drug sensitivity. Taking these findings together, NECTIN4 is suggested to play roles in the progression of melanoma, presumably through activation of the PI3K/Akt signaling pathway.

In this report, we used two drug-resistant cell lines and drug-resistant SKMEL28 cells seemed to be more susceptible to the treatments than drug-resistant MEL1617 cells. Although these cell lines are both derived from cutaneous melanoma bearing BRAF^V600E^ mutation, they are known to have different genetic statuses regarding *PTEN*, *P53*, *CDKN2A*, and *CDK4* as reported by Villanueva and colleagues [43]. MEL1617 has wild type *PTEN*, *P53*, *CDK4*, and lacks *CDKN2A*. On the other hand, SKMEL28 has mutated *PTEN* and *CDK4*, wild type *CDKN2A*, and lacks *P53*. To our knowledge, direct relationships between NECTIN4 and *PTEN*, *P53*, *CDK4*, or *CDKN2A* have not been reported, but differences in these genes may affect the susceptibility of cells to the treatments.

Resistance to BRAFi is a serious concern for the treatment of unresectable melanoma. Although BRAFi improved the survival of patients with BRAF-mutated melanoma, its effect does not last long (a median time to progression of 5.1–8.8 months) [34] due to the acquisition of drug resistance with multiple causes [34,35,36,37]. In addition, less than 20% of patients with BRAF-mutated melanoma present intrinsic resistance and do not respond to BRAFi [14,36,44]. To overcome these issues, the combination of BRAFi/MEKi is now used in a clinical setting, and it significantly improved the response rate and duration [34,45]. ICI has also improved the prognosis of melanoma patients [6,7]. These therapies constitute a major change in the treatment of advanced melanoma; however, a subset of patients does not respond or still develops drug resistance [8,9,34]. Novel drugs effective for BRAFi/MEKi-resistant or ICI-resistant melanoma are thus desired. Although the association between BRAFi resistance and NECTIN4 expression has not been reported, we found that the expression of NECTIN4 was increased, along with the acquisition of BRAFi resistance in melanoma cells. In addition, ERK signal activation, one of the mechanisms of BRAFi-resistance acquisition and observed in our DR cell lines, was slightly but significantly inhibited by NECTIN4 knockdown (Figure 6). Although detailed investigation is required, NECTIN4 may partly contribute to the acquisition of drug-resistance and these results imply the potency of NECTIN4 as a target of novel drugs for BRAFi-resistant melanoma.

Taking our findings together, we have shown that 1 NECTIN4 is expressed in melanoma patients and in melanoma cells, 2 NECTIN4 is upregulated in response to the acquisition of BRAFi resistance, and 3 melanoma cells are sensitive to MMAE (an anti-mitotic drug that is a component of ADCs). Since the phenotypes and the results observed in this research are based on the short-term culture and reflect transient phenomena, further long-term experiments need to be performed in the future. Even though our results suggest that the NECTIN4-targeting ADCs, such as enfortumab vedotin, exert antitumor effects against melanoma regardless of their BRAF mutation status, even after the melanoma cells develop to acquire BRAFi resistance.

## 4. Materials and Methods 

### 4.1. Cell Culture

MEL1617 and SKMEL28 melanoma cell lines (ATCC, Rockville, MD, USA) were cultured in basal media: RPMI1640 (Sigma-Aldrich, St. Louis, MO, USA; R8758) supplemented with 10% fetal bovine serum (Nichirei Biosciences, Tokyo, Japan; 174012), and 0.1% DMSO (Sigma-Aldrich; 07-4860-5). DR melanoma cells were maintained in basal media with 1 µM dabrafenib (ChemScene, Deerpark, NJ, USA; CS-0692). Cells were passaged at a sub-confluent and media were refreshed every 2 days. Cells were used within five passages after thaw and verified for correct morphology by microscopic observation. Cell images were captured with a microscope (Nikon Corporation, Tokyo, Japan; Nikon ECLIPSE TS100).

### 4.2. Generation of BRAFi-Resistant Melanoma Cells

Cells were chronically treated with stepwise increased concentrations of dabrafenib. Cells were selected at each step until they resumed the same growth kinetics as the untreated parental line before moving to the next step. Throughout the 6 weeks of treatment, the dabrafenib concentration was increased from 0.3 to 1.0 µM. The sensitivity of cells to dabrafenib was assessed using MTS-based cell viability assays.

### 4.3. Cell Viability 

Cell viability was determined using Cell Counting Kit-8 (CCK-8, Dojindo, Kumamoto, Japan; 347-07621). Cells were seeded in 96-well plates (2000–5000 cells/well), treated with MMAE (1.25, 2.5, 5.0, 7.5, and 10 nM, ChemScene; CS-0837), siRNAs, or an Akt inhibitor LY294002 (10 µM, Abcam, Cambridge, UK; ab120243), and incubated at 37 °C with 5% CO_2_. After 1–3 days, cells were treated with CCK-8 solution for 2–4 h and absorbance at 450 nm was measured using an iMark microplate reader (Bio-Rad Laboratories, Hercules, CA, USA; 1681130J1).

### 4.4. siRNA Transfection

Cells were transfected with control siRNA (Invitrogen, Carlsbad, CA, USA; AM4611) or NECTIN4 siRNA (Invitrogen; s37689) using Lipofectamine RNAiMAX (Invitrogen; 13778075). Briefly, siRNAs were diluted with Opti-MEM^TM^ I Reduced Serum Medium (Thermo Fisher Scientific, Waltham, MA, USA; 31985062), mixed with Lipofectamine, and incubated for 20 min at room temperature. Then, the siRNA-Lipofectamine complexes were mixed with the cells (final siRNA concentration of 10 nM). At 24–72 h after transfection, the cells were harvested and used for further analysis. The knockdown efficiency was determined by quantitative reverse-transcription polymerase chain reaction (qRT-PCR) and Western blotting. 

### 4.5. RNA Extraction and qRT-PCR

RNA was extracted using RNeasy Mini Kit (Qiagen, Hilden, Germany; 74014), converted to cDNA by PrimeScript RT Reagent Kit (Takara Bio Inc., Kusatsu, Japan; RR037), and used for PCR with TB Green Premix ExTaq (Takara Bio Inc.; RR420) and CFX Connect^TM^ Real-Time System (Bio-Rad Laboratories; 1855201J1). The sequences of primers were as follows: *NECTIN4* forward; 5′-CAAAATCTGTGGCACATTGG-3′, reverse; 5′-GCTGACATGGCAGACGTAGA-3′, and β-Actin (*ACTB*, an internal control gene) forward; 5′-ATTGCCGACAGGATGCAGA-3′, reverse; 5′-GAGTACTTGCGCTCAGGAGGA-3′. The expression of each mRNA relative to that of control samples was calculated by the comparative Ct method.

### 4.6. Western Blotting

Western blotting was performed as reported previously [46]. The antibodies used were as follows: rabbit anti-human NECTIN4 (1:1000, Abcam; ab192033), rabbit anti-human Akt (1:1000, #9272), rabbit anti-human phospho-Akt (1:2000, #4060), rabbit anti-human ERK (1:1000, #9102), rabbit anti-human phospho-ERK (1:2000, #4370), rabbit anti-human MEK (1:1000, #9122), rabbit anti-human phospho-MEK (1:1000, #9154), rabbit anti-human EGFR (1:1000, #4267), rabbit anti-human β-actin (1:2000, #4970), and goat anti-rabbit IgG horseradish peroxidase-linked secondary antibody (1:10,000, #7074) (all from Cell Signaling Technologies, Danvers, MA, USA). Immunological bands were visualized with SuperSignal^TM^ West Pico Chemiluminescence Substrate (Thermo Fisher Scientific; 34580) and captured with the ChemiDoc^TM^ XRS Plus System (Bio-Rad Laboratories; 1708265J1PC). The signals of bands were measured with Image Lab Software (Bio-Rad Laboratories). 

### 4.7. Annexin V-PI Staining and Flow Cytometry

Cells were seeded in 6-well plates and transfected with siRNAs as mentioned above. At 48 h, cells were harvested by pipetting and labeled with Annexin V-FITC and PI in accordance with the instructions of the Annexin V-FITC Apoptosis Detection Kit (Nacalai Tesque, Kyoto, Japan; 15342-54). Apoptotic cells were detected with a FACSCanto II flow cytometer (BD Biosciences, Franklin Lakes, NJ, USA) and analyzed with FlowJo software (Tree Star, San Carlos, CA, USA). 

### 4.8. Patients

This study is a retrospective review of our patients and conducted in accordance with the guidelines of the Declaration of Helsinki. Immunohistochemical analysis using patient samples was approved by the Ethics Committee of Kyushu University Hospital (Approval ID: 30-363). Written informed consent was received from the patients prior to their inclusion in the study. We retrieved 126 patients with primary melanoma lesions who were treated at the Department of Dermatology of Kyushu University, Fukuoka, Japan, between July 2001 and March 2019. At least three experienced dermatopathologists confirmed the diagnosis. Clinical and demographic data of patients were collected from the patients’ files and analyzed.

### 4.9. Immunohistological Analysis

All formalin-fixed paraffin-embedded melanoma tissues were obtained from the archives of Kyushu University Hospital. Immunohistochemistry staining was performed as reported previously with slight modifications [47,48]. Sections were incubated with following primary antibodies; rabbit anti-human NECTIN4 (1:150, Abcam; ab192033) for 30 min or with mouse anti-human VE1 (1:100, Abcam; ab228461) for 90 min at room temperature. After incubation with N-Histofine Simple Stain AP MULTI (Nichirei Biosciences; 414261) secondary antibody for 30 min, sections were treated with FastRed II (Nichirei Biosciences; 415261) and counterstained with hematoxylin. 

For immuno-double staining, antigen retrieval was performed with Heat Processor Solution pH9 (Nichirei Biosciences; 715291) at 100 °C for 40 min. Sections were then incubated with mouse anti-human SOX10 (prediluted, Nichirei Biosciences; 418241) for 30 min at room temperature. N-Histofine Simple Stain AP MULTI and Perma Blue Plus/AP (Diagnostic BioSystems, Inc., Pleasanton, CA, USA; K058) were used as a secondary antibody and chromogen, respectively. After endogenous peroxidase blockade with 3% H_2_O_2_ (Nichirei Biosciences), sections were incubated with rabbit anti-human NECTIN4 for 30 min and further treated with N-Histofine Simple Stain MAX-PO MULTI (Nichirei Biosciences; 724152) secondary antibody for 30 min at room temperature. Immunoreactions were detected by using 3,3′-diaminobenzidine tetrahydrochloride (Nichirei Biosciences, 725191) as a chromogenic substrate. In this method, SOX10- and NECTIN4-positive signals are shown in blue and in brown, respectively.

### 4.10. Evaluation of NECTIN4 Immunohistochemical Staining

NECTIN4 expression in patients’ melanoma tissue was evaluated by a semiquantitative method utilizing H-score [33]. NECTIN4 staining intensity was classified as follows: no staining (0), weakly positive (1+), moderately positive (2+), and strongly positive (3+) (Figure 1a). The staining of NECTIN4 in epidermis was categorized as 2+ and considered as an internal control. The H-score was calculated by multiplying the percentage of NECTIN4-positive cells and a staining intensity ranging from 0–3+. Regarding VE1, tissues were considered as VE1-positive when at least 5% of melanoma cells were stained for VE1 according to the previous reports [47,48]. Two dermatologists (M.M. and T.I.), who were blinded to the clinical information, independently assessed NECTIN4 expression in the sections. Images were captured using a microscope (Nikon Corporation; ECLIPSE 80i). 

### 4.11. Statistics

Quantitative results from at least three experiments are indicated as mean ± standard deviation. GraphPad Prism7 software (GraphPad Software, San Diego, CA, USA) was used for statistical analyses. The differences between two groups were assessed by Student’s unpaired two-tailed *t*-test. Fisher’s exact test was applied to analyze the relationship between two variables. Patients’ survival (MSS, DFS, and OS) was calculated using the Kaplan–Meier method and the log-rank test. For multivariate survival analysis, the multivariate Cox proportional hazards regression was applied. *p*-value < 0.05 was considered as statistical significance. 

## 5. Conclusions

In conclusion, we have investigated the expression and roles of NECTIN4 in melanoma. NECTIN4 was shown to be frequently expressed in BRAF-mutated melanoma. NECTIN4-targeted therapy with ADCs, such as enfortumab vedotin, is a potential candidate for treating melanoma with BRAF-mutation and BRAFi-resistant melanomas, and its efficacy needs to be assessed in future research.

## Figures and Tables

**Figure 1 ijms-22-00976-f001:**
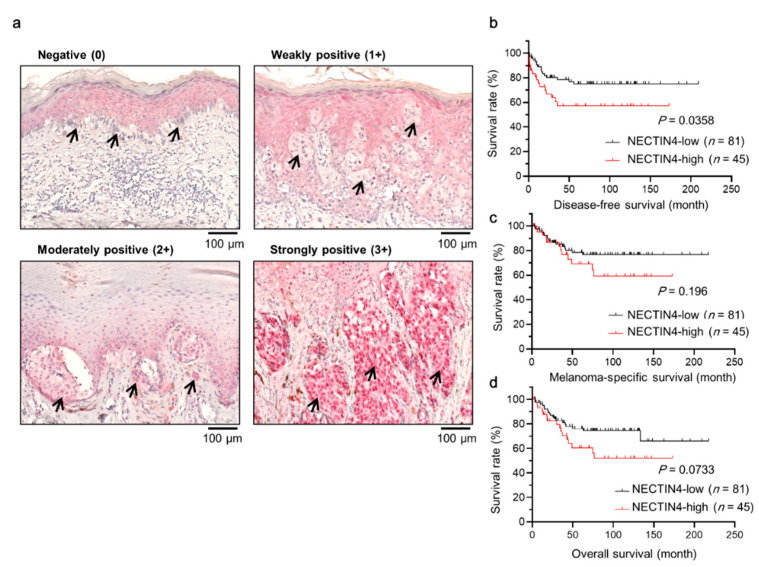
NECTIN4 is expressed in melanoma tissues and is associated with BRAF mutation. Immunohistochemical staining of NECTIN4 in a patient’s tumor tissue. (**a**) Representative images from 126 melanoma samples are shown. The intensity of the staining was classified on a scale from 0 to 3+: no staining (0), weakly positive (1+), moderately positive (2+), and strongly positive (3+). Black arrows indicate NECTIN4-positive cells. Scale bar = 100 µm. (**b**–**d**) Survival curve of NECTIN4 expression intensity and (**b**) disease-free survival, (**c**) melanoma-specific survival, and (**d**) overall survival (*n* = 126). Patients’ survival was calculated using the Kaplan–Meier method and the log-rank test. *p* < 0.05 was considered to indicate a statistically significant difference.

**Figure 2 ijms-22-00976-f002:**
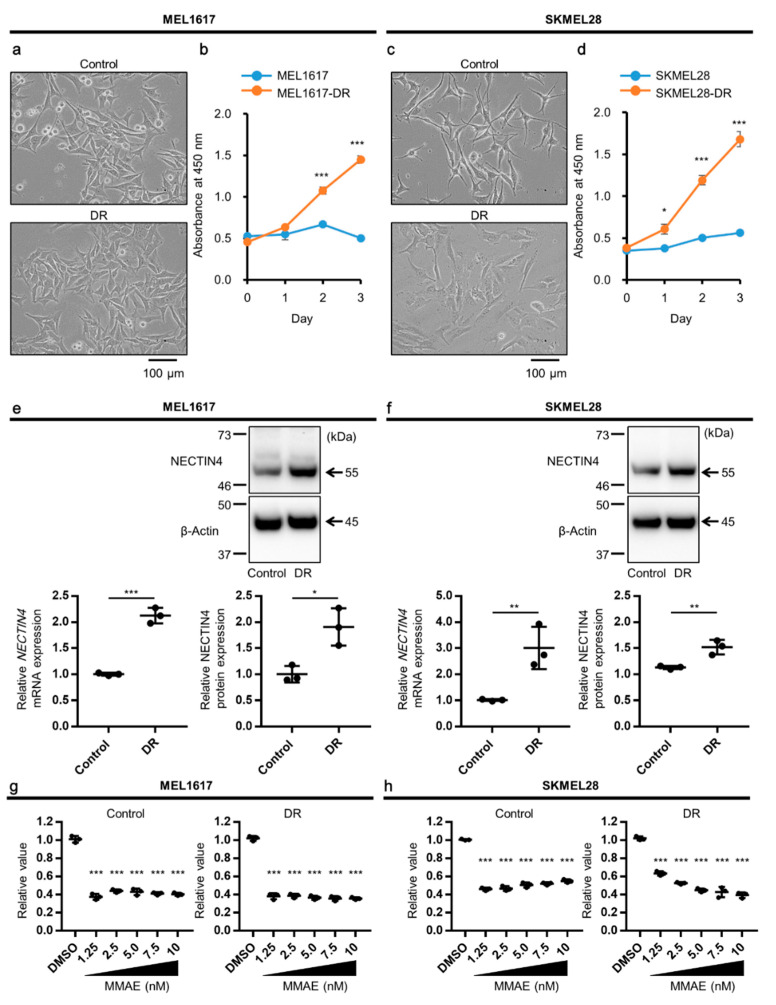
BRAFi-resistant melanoma cells highly express NECTIN4 and are sensitive to MMAE. (**a**) Representative images of MEL1617 (Control) and its dabrafenib-resistant (DR) cells. Scale bar = 100 µm. (**b**) Growth curve of MEL1617 and MEL1617-DR treated with 1 µM dabrafenib. (**c**) Representative images of SKMEL28 (Control) and its DR cell lines. Scale bar = 100 µm. (**d**) Growth curve of SKMEL28 and SKMEL28-DR treated with 1 µM dabrafenib. (**e**,**f**) The expression of *NECTIN4* mRNA and NECTIN4 protein in (**e**) MEL1617 and (**f**) SKMEL28 and their DR cell lines (*n* = 3). Representative gel images of NECTIN4 and β-Actin (a loading control) are shown. (**g**) MEL1617 and (**h**) SKMEL28, and their DR cell lines were treated with DMSO (0.1%) or MMAE (1.25, 2.5, 5, 7.5, or 10 nM) for 48 h and assessed for cell viability (*n* = 3). Data are presented as the mean ± standard deviation of three independent experiments. * *p* < 0.05, ** *p* < 0.01, and *** *p* < 0.001 determined by Student’s unpaired two-tailed *t*-test.

**Figure 3 ijms-22-00976-f003:**
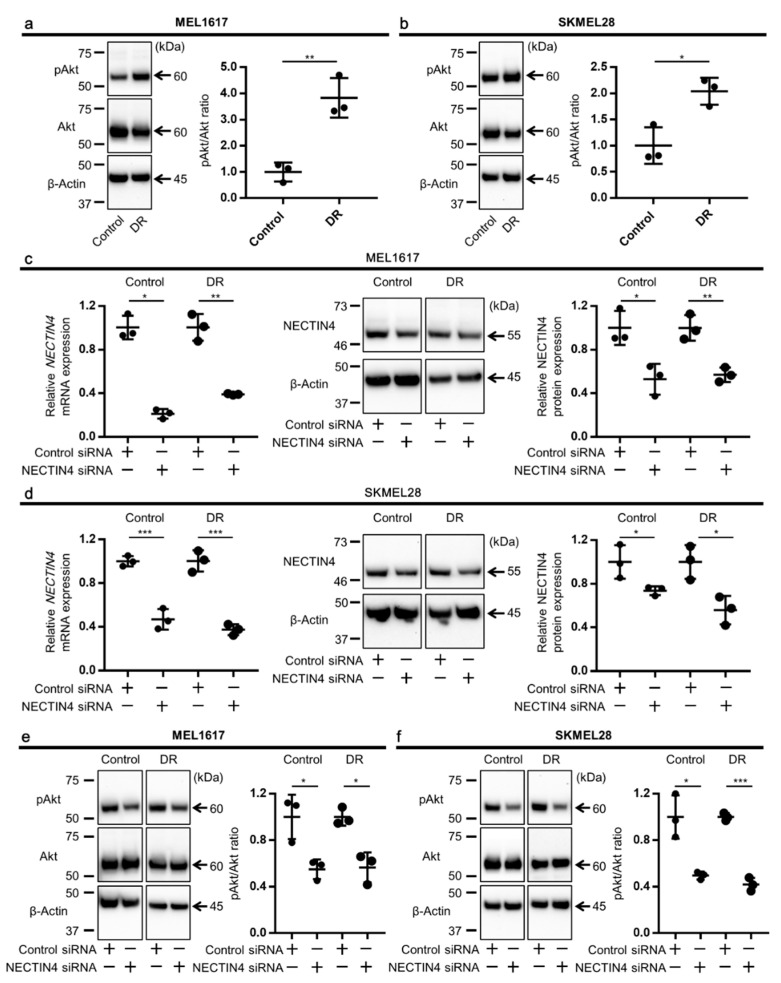
Phosphorylation of Akt is upregulated in BRAFi-resistant melanoma cells which are inhibited by NECTIN4 silencing. The pAkt/Akt ratio was determined by Western blotting. Representative gel images and relative pAkt/Akt ratios in (**a**) MEL1617 and (**b**) SKMEL28, and their DR cell lines are shown (*n* = 3). Cells were transfected with control or NECTIN4 siRNA for 48 h. Knockdown efficiency of NECTIN4 at mRNA (left panels) and at protein (middle and right panels) in (**c**) MEL1617 and (**d**) SKMEL28, and their DR cell lines are shown. pAkt/Akt ratio in (**e**) MEL1617 and (**f**) SKMEL28, and their DR cell lines transfected with NECTIN4 siRNA for 48 h. Representative gel images and the pAkt/Akt ratio relative to that of control siRNA-treated samples are shown (*n* = 3). Data are presented as the mean ± standard deviation of three independent experiments. β-Actin served as a loading control. * *p* < 0.05, ** *p* < 0.01, and *** *p* < 0.001 determined by Student’s unpaired two-tailed *t*-test.

**Figure 4 ijms-22-00976-f004:**
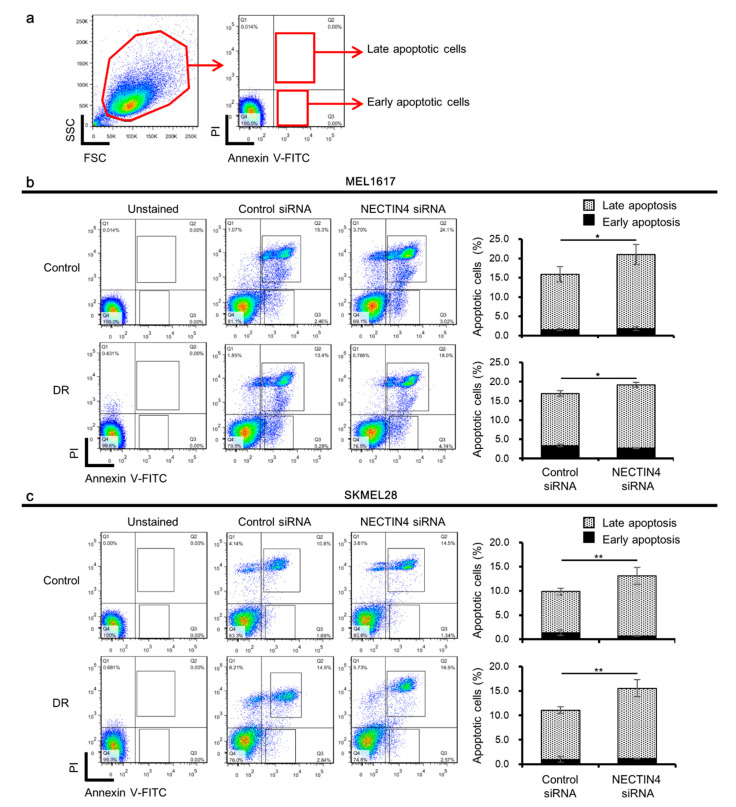
NECTIN4 is involved in the regulation of apoptosis in BRAFi-resistant melanoma cells. (**a**) Gating strategy of flow cytometric analysis. Non-specific signals were removed by the first gate. The gated fraction was further expanded with Annexin V-FITC and PI, and analyzed for the percentage of late (Annexin V-positive PI-positive) and early (Annexin V-positive PI-negative) apoptotic cells. Representative flow cytometric images (left panels) and the percentage of apoptotic cells among all cells (right panels) of (**b**) MEL1617 and (**c**) SKMEL28, and their DR cell lines are shown (*n* = 3). Data are presented as the mean ± standard deviation of three independent experiments. * *p* < 0.05 and ** *p* < 0.01 determined by Student’s unpaired two-tailed *t*-test.

**Figure 5 ijms-22-00976-f005:**
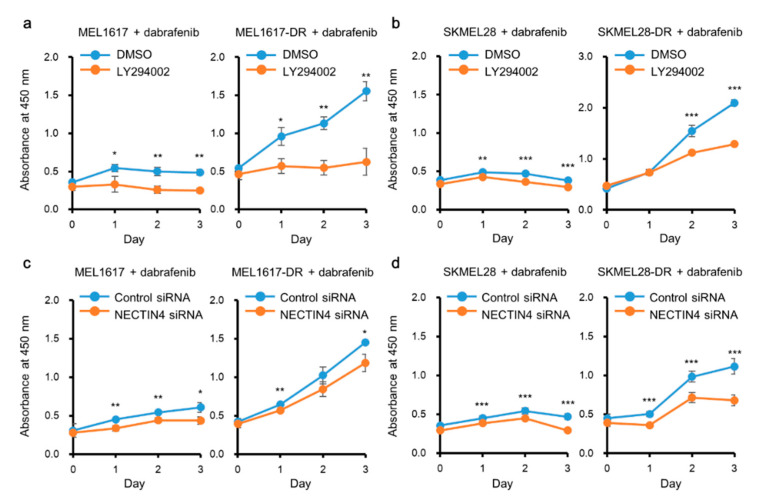
NECTIN4 inhibition increases sensitivity of BRAFi-resistant melanoma cells to BRAFi. (**a**) MEL1617 and (**b**) SKMEL28, and their DR cell lines were treated with DMSO (0.1%) or PI3K inhibitor LY294002 (10 µM) with dabrafenib for 3 days. (**c**) MEL1617 and (**d**) SKMEL28, and their DR cell lines were transfected with control or NECTIN4 siRNA in the presence of dabrafenib for 3 days. Viable cells in each condition were detected using Cell Counting Kit-8 (CCK-8). Experiments were performed in triplicate wells and repeated three times (*n* = 9). Data are presented as the mean ± standard deviation. * *p* < 0.05, ** *p* < 0.01, and *** *p* < 0.001 determined by Student’s unpaired two-tailed *t*-test.

**Figure 6 ijms-22-00976-f006:**
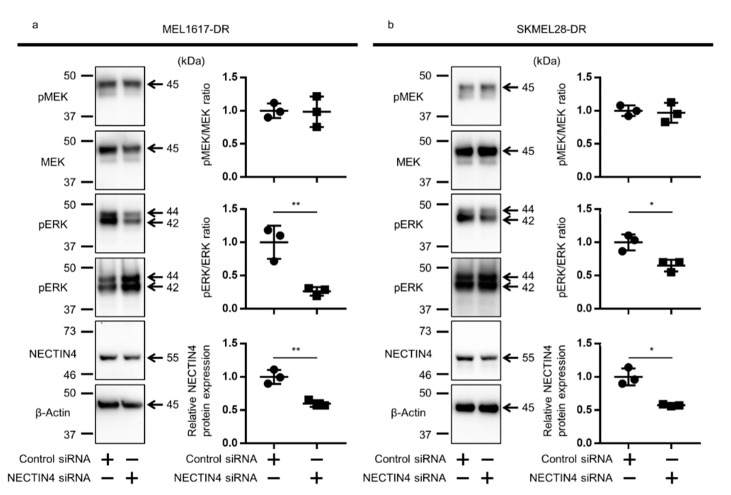
NECTIN4 inhibition downregulates phosphorylated ERK in MEL1617-DR and SKMEL28-DR cell lines. (**a**) MEL1617-DR and (**b**) SKMEL28-DR cell lines were transfected with control or NECTIN4 siRNA and assessed for pMEK/MEK and pERK/ERK ratio. Representative blot images (left panels), and pMEK/MEK ratio, pERK/ERK ratio, and relative NECTIN4 expression (right panels, *n* = 3) are shown. Data are presented as the mean ± standard deviation of three independent experiments. β-Actin served as a loading control. * *p* < 0.05 and ** *p* < 0.01 determined by Student’s unpaired two-tailed *t*-test.

**Table 1 ijms-22-00976-t001:** Basic demographic and clinical characteristic data of all 126 patients with primary malignant melanoma.

Parameters		Number
Age	Mean	63.7
	Median	69
	Range	16–88
Sex	Male	55 (43.7%)
	Female	71 (56.3%)
Tumor site	Head and neck	12 (9.5%)
	Trunk	5 (4.0%)
	Upper limb	3 (2.4%)
	Hand	25 (19.8%)
	Lower limb	7 (5.6%)
	Foot	74 (58.7%)
Histopathological subtype	ALM	94 (74.6%)
	SSM	15 (11.9%)
	LLM	10 (7.9%)
	NM	7 (5.6%)
Ulceration	Present	52 (41.3%)
	Absent	70 (55.6%)
	Unknown	4 (3.2%)
T category	Tis	29 (23.0%)
	T1	20 (15.9%)
	T2	13 (10.3%)
	T3	21 (16.7%)
	T4	41 (32.5%)
	Unknown	2 (1.6%)
N category	N0	85 (67.5%)
	N1	15 (11.9%)
	N2	11 (8.7%)
	N3	14 (11.1%)
	Unknown	1 (0.8%)
M category	M0	115 (91.3%)
	M1	11 (8.7%)
VE1 staining	Positive	38 (30.2%)
	Negative	88 (69.8%)

Data are presented as *n* (%). ALM, acral lentiginous melanoma; SSM, superficial spreading melanoma; NM, nodular melanoma; LLM, lentigo malignant melanoma.

**Table 2 ijms-22-00976-t002:** Factors associated with NECTIN4 expression.

Parameter		NECTIN4 Expression		*p*-Value
		Low	High	
Age	<70	46	25	>0.999
	≥70	35	20	
Sex	Male	35	20	>0.999
	Female	46	25	
Histopathological subtype	ALM	62	32	0.527
	Others	19	13	
Ulceration	Present	33	19	0.849
	Absent	46	24	
T category	Tis, T1, T2	39	20	>0.999
	T3, T4	42	23	
N category	N0	55	30	0.843
	N1–3	25	15	
M category	M0	74	41	>0.999
	M1	7	4	
VE1 staining	Positive	16	22	0.0011 *
	Negative	65	23	

Data are presented as *n* (%). * *p* < 0.05 indicated statistical significance determined by the multivariate Cox proportional hazards regression analysis. ALM, acral lentiginous melanoma.

## Data Availability

The data presented in this study are available in the main text and the Appendix A.

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
