# Peer review of "NECTIN4: A Novel Therapeutic Target for Melanoma"

_ijms, 2021, doi:10.3390/ijms22020976_

Round 1

Reviewer 1 Report

Well done, thanks for the comprehensive revision!

Reviewer 2 Report

The revised manuscript regarding the use of NESTIN4 as a novel target for melanoma treatment is significantly improved compared to the previous version and all the results presented are reasonable.

I would suggest moving supplementary figure 7, or at least some parts of it into the main body of a manuscript. In terms of combating drug resistance of melanoma, the decrease in ERK1/2 phosphorylation after nectin4 knockdown is a very important result.

Figure 2 shows that drug-resistant cells are characterized with increased nectin4 expression when compared to drug-naïve cells, which is confirmed at both mRNA and protein levels (e and f), whereas in Figure 3 we see no difference in nectin4 expression between drug-naïve and drug-resistant cell lines (c and d). Can the authors explain this discrepancy?

All the results in the manuscript indicate that SKMEL28 dabrafenib-resistant cell line is significantly more susceptible to cell death after nectin4 knockdown, when compared with dabrafenib-resistant MEL1617 cell line. Did the authors try to discuss that? What are the genetic differences between these two cell lines? It would be good to include these presumptions in the discussion part of the manuscript.

There are several minor language mistakes, such as plural/singular forms, that need to be corrected as well.

Reviewer 3 Report

The authors have mainly addressed most of my concerns and the mauscript has been improved. although some key experiments to prove the specificity are missing, the data overall look convinving. The main points of specificity is respect to generate resistant cell lines by using BRAFi. It is surprising that the authors claim that resistance occurs shorty after introducing the drug. Dabrafenib causes cytostatic effects to the cells for a period of time in vitro and in vivo and after that cells start to acquire resistance after the reactivation of MAPK pathway. In this study it seems that cells grow without having any effect the inhbition of proliferation and it seems that are already resistant. is this the case? Second, to prove the specificity of the inhibition and the generation of resistant cell lines, a non Braf cell line could have been used (e.g. Nras). Last concern is the fact that siRNA doesn't lead to protein decrease...is that expected? Did the authors validate or investigated this further? This is also crucial for the study. If the protein is not depleted then it can still be impliocated in the MAPK pathway.  Finally as all the experiemental step is based on short term culture experiments I would  reform the conclusions and be more careful stating that the phenotype and the outcome is transient and further long term experiments need to be done in the future to prove their hypothesis.

Author Response

This manuscript is a resubmission of an earlier submission. The following is a list of the peer review reports and author responses from that submission.

Round 1

Reviewer 1 Report

Dear authors,

this is an absolut interesting and well-presented manuscript which provides novel findings to the readership. However, some points needs to be adressed prior publication. Please find my comments below:

  • Authors have investigated the levels of Nectin4 expression particularly in acral lentigineous melanoma, as shown in Figures 1 and S1. However, a second marker is missing that a) shows the melanoma origin, here it would be intersting to see if there is an overlap of typical melanoma markers e.g. MART1 or if Nectin4 expression is rather pointing out a stem-like phenotype. B) Do acral lentigineous melanoma predominantly show Nectin4 expression, which would be interesting to know as acral lentigineous areas are not necessarily the sun exposed areas. So, is there any knowledge about sun exposure and NEctin4 expression?
  • Authors should discriminate o rat least show whether there is a significant difference in Nectin4 expression in BRAFwt vs BRAFmut melanoma
  • Is there any evidence that Nectin4 expression predominantly labels primary and non-metastatic melanoma? So, is Nectin4 expression associated with normal melanoma progression or rather associated with resistance?
  • In Figure 1 b-c: Did you check out cBioPrtal for Nectin4 expression of melanoma? This might be a good comparision, how Nectin4 expression related with overall-survival?
  • Figures 2a-b: I would recommend the conversion of images to gray-scales which would remove the blueish background.
  • A general comment for western blots: please state the correct band location with an arrow on the right side of the blot, this makes it more convenient fort he readers. Although the diffrences in mRNA and protein levels of Nectin4 are quite obvious, it is hard to see differences in protein levels in the siRNA-mediated knockdown, please include a quantification of bands of your western blots. This would be very helpful.
  • Figure 3: Levels of Nectin4 must be shown in the main figure even when the knockdown is not that obvious. Either repeat the experiment with a higher dose of siRNAs or directly establish a stable shRNA-based system which might be more promising of use advantage of CRISPR/Cas9 if possible.
  • Figure 4: If cell with a knockdown of Nectin4 show a strong apoptosis, do authors have included these cells fort he preparation of protein lysates? If dead cells are simply lost due to the siNectin4-mediated apoptosis they are probably missed, hence in the western blots only Nectin4 levels of remaining and viable cells is shown. As the effects oft he Nectin4 knockdown are very clear, this might be the case. Hence, an even stronger knockdown might be associated with enhanced apoptosis.
  • Figure 5: Does an overexpression of Nectin4 would mediate resistance in these cell lines? In Figure 5d-f it seems as there is only a minor effect of siRNA but maybe the same problem as mentioned in Figure 4 is present, siNectin4 cells undergo apoptosis, hence are not present/attached anymore during the treatment.

Reviewer 2 Report

The manuscript regarding the use of NESTIN4 as a novel target for melanoma treatment is well prepared and well written, however I have several doubts concerning the data presented. First of all, it seems that the vemurafenib-resistant melanoma cell lines did not acquire the resistance to vemurafenib:

  1. The Authors mentioned that „the shape of VR cells was similar to that of the control cells (fig 2a)”, whereas vemurafenib-resistant melanoma cell lines are characterized by altered phenotype visible under the microscope, when compared to sensitive cells;
  2. There is very slight increase in NECTIN4 protein levels in vemurafenib-resistant cells vs control cells, when compared to dabrafenib-resistant cells vs control cells (fig 2c),
  3. The increase in pAKT/AKT protein levels are not statistically significant, when compared to control cells (fig 3c)
  4. Virtually no visible difference in apoptosis rate in vemurafenib-resistant cells vs control (fig 4b)
  5. The viability of drug-naive MEL1617 cell lines is exactly the same as the viability of vemurafenib resistant cells grown in the presence of this drug (fig 5c and fig 5e, even after NECTIN4 knockdown). It is practically impossible that for some points in the fig 5cdef the p-value is < 0.01, or even <0.001. The statistics computations should be re-checked for the whole manuscript.
  6. Why vemurafenib-resistant cell lines were not prepared for SKMEL28 cell lines?

In conclusion, it seems that the „vemurafenib-resistant” MEL1617 cell lines behave exactly like drug-naive parental cells. The Authors should include some additional results that would confirm the acquisition of the resistance to both vemurafenib and dabrafenib (eg. decreased proliferation, reactivation of MEK/ERK pathways and increase in EGFR expression in drug-resistant vs drug-naive cells). For now, the best solution is to get rid of vemurafenib data and focus on dabrafenib results, which are slightly better.

Other issues:

  1. I don’t understand the meaning of single and double asterisks in figure 2e and 2f. What are they refering to?
  2. The decrease in viability of both drug-naive and drug resistant is practically the same, nevertheless NECTIN4 expression levels. That might suggest that drug resistance is not influenced by the increase in NECTIN4 expression.
  3. The decrease in akt phosphorylation is visible even in drug-naive cells and the level of decrease is similar to that in drug-resistant cells (fig 3cd). That might suggest that drug resistance is not influenced by the NECTIN4 level.
  4. As mentioned above, there is slight or no difference in apoptosis after NECTIN4 knockdown in MEL1617 cell lines, both drug-resistant and –sensitive (fig 4). The results for SKMEL28 are a bit better.
  5. The efficacy of siRNA-mediated knockdown on protein level should be moved to the main body of manuscript from supplementary data and compared with the p-Akt/Akt expression in figure 3. Besides, the efficacy of siRNA-mediated knockdown after 48 h is very low (supplementary figure S2), but clearly increases after 72 h for SKMEL28 cells (supplementary figure S6). Why the Authors did no chose this time point of siRNA-mediated knockdown for the subsequent experiments?

Reviewer 3 Report

In this study, Tanaka et al., provide the clinical relevance of NECTIN4 in melanoma and present data as a potential therapeutic target to overcome resistance upon targeted therapy. In addition they provide a link of NEctin 4 with PI3K pathway activation. 

Overall the study looks novel and solid with some strong points, especially the use of NEctin 4 as biomarker and/or potential target to suppress tumor growth. 

However there are few concerns raised and need further support and experimental data. 

  1. The authors claim correlation of Nectin 4 with BRAF. Is there any correlation with Nras mut, NF1 and 3WT? What is the status of NEctin 4 in patients with these mutations. Is overall survival significantly prolonged in patients with low Nectin 4?  What is the expression of Nectin 4 in primary vs metastatic biopsies. Did the authors mined publicly available databases (TCGA, CCLE).
  2. the correlation of Nectin 4 with BRAF positivity is not shown (line 127).
  3. How the authors define resistant cell lines unsing Brafi. What is the status of pERK in the restistant cells. Importantly, since the standard of care in melanoma regarding to targeted therapy is Dabrafenib and Trametinib that leads to better inhibition of MAPK pathway why do they apply only BRAFi. Would the effect on cells and on NEctin 4 expression be more profound?
  4. does nectin 4 inhbition in Restistance cell lines decreases the MAPK pathway activation?
  5. what is the effect of NEctin expression in NRAS mut cell lines blocking MAPK pathway through MEK inhibitors? The outcome of this will further confirm the selective expression of Nectin 4 in BRAF mut melanoma.
  6. the genetic inactivation of Nectin 4 at protein level doesn't look convinving enough. Do the authors have a possible explanation for this?
  7. Overall the experimental design is based on short term culture cells that the effect of inhbition either by siNectin or LY can be transient. The authors should address some of the key findings in long term culture conditions. 
  8. I feel like the introduction is too long and doesn't go directly to the point. It needs to be shortned and reformated.